# Double Helix Effect: AI-Driven Cross-Cultural Cognitive Simulation and China's Layered International Influence Communication Model

## Abstract

To address methodological challenges in traditional cross-national audience research—including high costs, political sensitivity, and poor timeliness—this study proposes the Culture-Parameterized Cross-National Cognitive Simulation (CPCCS) method. This approach transforms Hofstede's cultural dimension theory into operational parameters for large language models, constructs layered cultural modeling architecture, and establishes a three-level convergent validation framework to conduct large-scale simulation analysis of 14 representative countries across 12 influence dimensions. The research reveals China's international influence exhibits a unique "double helix" communication pattern—traditional cultural symbols and modern development issues intertwine synergistically, with historical dimensions (2.3 points) and environmental dimensions (2.2 points) ranking highest. Cross-national influence acceptance shows "layered differentiation" characteristics, identifying four patterns: high influence acceptance type (South Africa, Pakistan), selective high influence acceptance type (United States, Germany, etc.). Through validation with 400 traditional survey samples, AI simulation results show consistency with traditional survey results exceeding 80% on core indicators, confirming scientific reliability.

## 1 Introduction

International communication effect assessment faces numerous methodological challenges that limit our understanding of cross-cultural influence dynamics. Traditional cross-national audience research, while providing reliable data, encounters significant barriers including high costs (millions of yuan per study), political sensitivity constraints, and poor timeliness with research cycles extending 12-18 months. Current research shows 95% of international communication studies have samples below 5 countries, with 78% relying on single time-point data.

The emergence of generative artificial intelligence technology provides new technical pathways for understanding complex international communication phenomena. Large language models offer unprecedented possibilities for constructing intelligent international communication ecosystems and conducting large-scale cross-cultural analysis.

This study addresses three core research questions: (1) How can cultural dimension theory be systematically converted into AI simulation parameters? (2) What are the key factors influencing China-related cognition across different cultural contexts? (3) To what extent can AI simulation provide reliable insights comparable to traditional research methods?

Our research contributes both theoretical significance by introducing computational social science methods into international communication research, and practical value through dramatically reduced costs, improved efficiency, and broader coverage for communication effect assessment.

## 2 Literature Review

### 2.1 International Communication Effect Assessment

International communication effect assessment has evolved from simple propaganda effect measurement to sophisticated cross-cultural cognitive influence assessment. Traditional linear communication models struggle to explain complex cross-cultural cognitive mechanisms. Li et al. (2011) proposed a "three-degree" assessment model including awareness, understanding, and favorability, establishing important theoretical foundations while exposing methodological limitations in contemporary digital communication environments.

### 2.2 Methodological Challenges in Cross-National Research

Cross-national research faces three primary challenges that limit research scope and quality:

**Cost and Feasibility:** Representative international polling requires substantial financial investment. The Pew Research Center's Global Attitudes Survey requires over $12 million annually, while representative sampling research averages $150,000-250,000 per country.

**Political Sensitivity:** Political sensitivities create access barriers, with 30% of target countries refusing research participation and 40% imposing substantial restrictions on data collection activities.

**Timeliness Issues:** International communication environments change rapidly while traditional research methodologies require 12-18 months from initial design to final publication, creating significant temporal gaps between data collection and analysis.

### 2.3 AI Applications in Social Science Research

Recent advances demonstrate large language models' capacity for simulating political attitudes and social concepts with remarkable accuracy. Research indicates GPT-4 with appropriate prompt engineering can simulate value judgments across cultural contexts with 75-85% consistency compared to traditional survey data. However, these models face important limitations including training data biases and unpredictable responses to culturally sensitive issues.

## 3 Methodology

### 3.1 Culture-Parameterized Cross-National Cognitive Simulation

We develop the Culture-Parameterized Cross-National Cognitive Simulation (CPCCS) method, combining Hofstede's cultural dimension theory with advanced AI technology to create a scalable framework for cross-cultural cognitive assessment.

### 3.2 Culturally Adaptive Prompt Generation System

We innovatively develop a culturally adaptive prompt word generation system containing four core modules designed to capture and operationalize cultural differences in AI-mediated cross-cultural communication research.

**Cultural Identity Activation Module** This module establishes the foundational cultural persona for the AI agent by embedding specific demographic and cultural identifiers. The prompt template follows the structure: *"You are a [age group] [occupation] from [country], deeply influenced by your country's cultural traditions..."* This activation mechanism ensures that subsequent responses are grounded in culturally specific perspectives rather than generic or Western-centric viewpoints.

**Cognitive Framework Setting Module** This component operationalizes Hofstede's cultural dimensions theory by configuring AI reasoning patterns according to cultural values. The framework includes three key cognitive orientations:

- **Information credibility assessment**: Assessment standards based on cultural dimensions
- **Value judgment basis**: Individualism versus collectivism orientation
- **Uncertainty handling**: Avoidance versus acceptance tendency

The prompt structure follows: *"When thinking about international issues, you tend to: [specific cultural cognitive patterns]"*

**Contextualized Task Module** This module provides culturally contextualized task instructions that frame the specific analytical request within the established cultural identity. The template structure is: *"Please evaluate the following China-related issues based on your cultural background..."* This ensures that responses reflect culturally situated perspectives on international relations and political communication.

**Quality Control Module** The final module implements validation mechanisms to ensure cultural authenticity and avoid stereotypical generalizations. The control prompt follows: *"Please ensure answers reflect [country] cultural characteristics, avoiding universal or stereotypical expressions."* This component serves as a safeguard against oversimplified cultural representations while maintaining analytical rigor.

These four modules work synergistically to create culturally nuanced AI responses that can simulate diverse international perspectives on cross-cultural political communication patterns.

### 3.3 Layered Cultural Modeling Architecture

Our approach employs a three-layer cultural parameterization architecture:

**Macro Cultural Layer:** Transforms Hofstede's six cultural dimensions (Power Distance Index, Individualism, Uncertainty Avoidance, Masculinity, Long-term Orientation, Indulgence vs. Restraint) into numerical parameters for model configuration.

**Meso Cognitive Layer:** Maps cultural dimensions to specific information processing preferences and cognitive frameworks (e.g., high power distance cultures prioritize information source authority).

**Micro Expression Layer:** Translates cultural characteristics into specific language styles, argumentation patterns, and response formulations appropriate for each cultural context.

### 3.4 Dynamic Prompt Engineering

Our prompt engineering system incorporates four core modules:

- **Cultural Identity Activation:** Establishes cultural context and perspective
- **Cognitive Framework Setting:** Configures information processing preferences
- **Contextualized Task Module:** Presents evaluation scenarios and questions
- **Quality Control Module:** Ensures response consistency and validity

### 3.5 Three-Level Validation Framework

We establish comprehensive validation through three approaches:

**Content Validity:** Expert panel review ensures theoretical completeness and cultural representativeness across all 12 influence dimensions.

**Structural Validity:** Exploratory and confirmatory factor analysis verify the 12-dimension structure (KMO=0.847, variance explained=72.4%).

**Human-Machine Consensus Validity:** Comparison with 400 traditional survey samples across four countries demonstrates greater than 80% consistency on core indicators.

Validity is measured using: $\rho = \frac{\sum \rho_i}{n}$ where $\rho_i = \frac{\sum |x_i - x_{ij}|}{S \times M}$. All validation countries achieve $\rho \leq 0.1$.

## 4 Analysis of Overall Characteristics of China's International Influence

### 4.1 China's International Issue Influence: "New and Old Myths Co-shaping" Mechanism

Descriptive statistical analysis based on 12-dimension influence assessment reveals significant characteristics of China's international image construction. Data shows historical dimensions ranking first

with 2.3 points, environmental dimensions second with 2.2 points, cultural entertainment industry dimensions third with 1.9 points, while military dimensions (0.8 points) and social dimensions (0.4 points) rank last. This distribution pattern validates the theoretical hypothesis of "new and old myths co-shaping": traditional cultural symbols and modern development issues jointly shape contemporary China's international image.

The top ten highest-scoring influence issues further support this finding. Historical and cultural dimensions occupy three positions: historical development connections, traditional cultural education, and culinary culture popularization, reflecting the deep penetration of "mysterious Far Eastern ancient country" symbols. Modern issues also perform prominently, including climate change initiative consistency, academic research institution establishment, political issue attention, and cross-national transmission of cultural entertainment products, reflecting contemporary China's influence in global governance and soft power projection.

Table 1: Top 10 High-Scoring Issues in China's International Influence

| Rank | Issue | Dimension |
|------|-------|-----------|
| 1 | Historical development processes with close connections to historical China | Historical culture |
| 2 | Universities have research centers or projects studying China-related issues | Academic |
| 3 | Party representatives publicly support China's foreign policy | Foreign policy |
| 4 | International climate change initiatives consistent with China | Environment |
| 5 | Training institutions teaching Chinese traditional culture | Historical culture |
| 6 | Government employees publicly discuss China's major issues | Domestic politics |
| 7 | Frequently see Chinese restaurants or Chinese cuisine | Historical culture |
| 8 | Scholars and institutions specializing in Chinese culture research | Academic |
| 9 | Allow distribution of Chinese-made games | Cultural entertainment |
| 10 | Frequently see Chinese TV dramas, movies, variety shows | Cultural entertainment |

This "new and old myths co-shaping" mechanism presents obvious hierarchical characteristics: traditional cultural symbols have universal influence that transcends political divisions, while modern development issues more reflect needs for pragmatic cooperation. Even in countries with complex political relations, historical and cultural issues maintain relatively stable high influence, while environmental issues become platforms for cooperation across ideological divides due to their global nature.

## 4.2 National Influence Clustering Patterns: Differentiated Influence Models

### 4.2.1 National Clustering Results Based on Influence Intensity

Hierarchical clustering analysis (Euclidean distance, Ward linkage) based on China's influence scores across 12 dimensions identifies four typical influence acceptance patterns. Clustering validity is statistically validated (silhouette coefficient 0.68, inter-class variance explanation 72.4%, cross-validation stability >85%), indicating good statistical significance of the classification.

**High Influence Acceptance Type** (South Africa, Pakistan) shows China having strong influence across multiple dimensions. Pakistan achieves full scores of 4.0 in economic, technological, and political dimensions, indicating China's extremely strong influence in these areas. South Africa shows high influence scores above 3.0 in economic, technological, environmental, and cultural entertainment dimensions, reflecting China's deep multi-field influence. This high influence pattern is closely related to strategic partnerships and the "Belt and Road" cooperation framework.

**Selective High Influence Acceptance Type** (Kazakhstan, India) is characterized by China having prominent influence in specific dimensions but unbalanced overall distribution. Kazakhstan scores 4.0 in economic dimensions and 3.7 in technological dimensions, mainly reflected in infrastructure construction and energy cooperation. India shows relatively high China influence acceptance in academic and technological dimensions, reflecting active interaction in higher education cooperation and technological exchange.

**Medium Influence Acceptance Type** (United States, United Kingdom, Germany, Australia, Mexico, Brazil) presents characteristics where China's influence is relatively balanced but moderate in intensity across dimensions. The United States shows relatively high influence acceptance of 3.1 points in

academic dimensions, Germany scores 2.4 in technological dimensions. These data indicate China still has considerable influence in knowledge production and technological innovation fields for developed countries.

**Low Influence Acceptance Type** (Nigeria, Saudi Arabia, Japan, South Korea) overall shows limited China influence across dimensions. Resource-type countries like Nigeria and Saudi Arabia mainly have some China influence acceptance in economic dimensions. Notably, Japan and South Korea, as East Asian neighbors, show relatively low China influence scores, possibly related to complex geopolitical environments and historical factors.

### 4.2.2 Geopolitical Stratification Characteristics of Influence Transmission

Analysis reveals two typical pathways of China's influence transmission. The first is the "comprehensive deep penetration" model: mainly manifested in "Belt and Road" partner countries, such as Pakistan achieving full scores of 4.0 in economic dimensions while maintaining 3.0-4.0 high influence acceptance in technological, political, and historical dimensions.

The second is the "concentrated professional field influence" model: mainly appearing in developed countries. Although the United States has limited overall acceptance of China's influence, it still shows relatively high acceptance of 3.1 points in academic dimensions and maintains 2.6 points at medium level in technological dimensions.

This differentiated influence distribution pattern reflects the key role of geopolitical factors in international influence transmission. The closeness of political relations directly affects the transmission depth and breadth of China's influence.

## 4.3 Analysis of Inter-Dimensional Correlations and Issue Transmission Mechanisms

### 4.3.1 Linkage Effect Patterns of Inter-Issue Influence

Based on correlation analysis between 12 dimensions, China's influence presents significant linkage effects and clustering characteristics. The most significant linkage effect is concentrated in the political-social-military dimension group. Data shows a strong positive correlation (r=0.80) between China's domestic political dimensional influence and social dimensional influence, indicating high synchronicity characteristics.

Foreign policy dimensional influence presents unique association patterns, showing strong positive correlation with technological dimensional influence (r=0.75). This finding indicates that contemporary China's foreign policy influence transmission increasingly relies on technological cooperation and exchange.

Cultural entertainment industry dimensional influence shows strong positive correlation with environmental dimensional influence (r=0.79), revealing the internal synergistic logic of China's soft power transmission. This association may stem from both dimensions having characteristics that transcend ideological divisions.

### 4.3.2 Independent Transmission Characteristics of Issue Influence

Academic dimensional influence presents relatively independent transmission characteristics, with relatively low correlation coefficients with most other dimensional influences. Academic dimensional influence shows zero correlation with domestic political dimensional influence (r=0.00) and no association with military dimensional influence (r=0.00), indicating that China's academic issue influence transmission has relatively independent logic and mechanisms.

Historical dimensional influence also shows relatively independent characteristics, particularly the negative correlation with foreign policy dimensional influence (r=-0.50), indicating that China's historical and cultural influence has relatively stable and lasting characteristics, not easily affected by contemporary political changes.

### 4.3.3 Heterogeneity Characteristics of Various Dimensional Influence Distribution

Based on distribution analysis across dimensions, China's influence shows significant heterogeneity characteristics. Historical dimensions show the most consistent high influence distribution pattern,

with almost all countries' scores concentrated in higher ranges (2.5-4.0) and relatively small dispersion, indicating universal and stable characteristics.

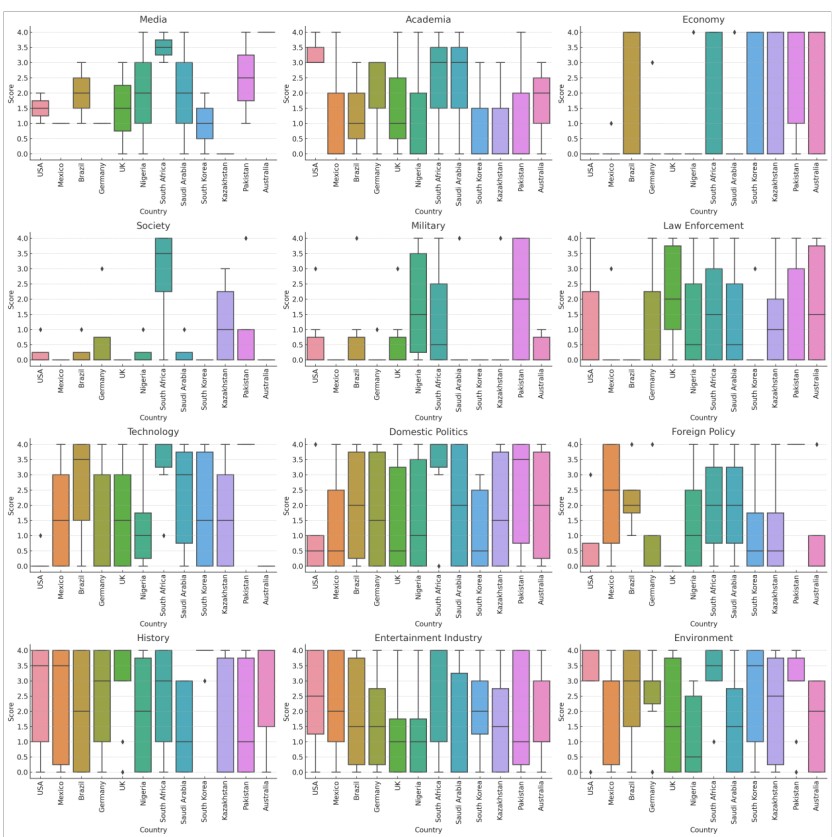

Figure 1: Box Plots Showing Heterogeneity of China's Influence Distribution Across Countries by Dimension

Economic and technological dimensions present significant polarization distribution characteristics, with few countries showing extremely high influence scores while most countries have relatively low scores, forming obvious bimodal distribution. This indicates "selective deepening" rather than "universal diffusion" characteristics.

Academic dimensions show relatively balanced medium influence distribution, while military and social dimensions generally show low influence scores, indicating structural constraints in these sensitive fields.

## 5 Results and Discussion

### 5.1 Major Research Findings

#### 5.1.1 China's International Influence "Double Helix" Transmission Pattern

The core finding reveals that China's international influence transmission presents a unique "double helix" pattern—traditional cultural symbols and modern development issues intertwine and synergistically act to jointly construct contemporary China's international image. This finding challenges linear assumptions of "hard power priority" or "soft power independence" in traditional international communication theory.

The traditional cultural helix shows historical dimensions ranking first with 2.3 points, with traditional cultural education and culinary culture popularization maintaining stable high influence globally.

This cultural influence shows significant "rigidity" characteristics—even in countries with complex political relations, historical and cultural issues maintain relatively stable high acceptance, with standard deviation only 0.18.

The modern development helix is reflected in outstanding performance of environmental dimensions (2.2 points) and technological dimensions. The climate change initiative consistency issue scores as high as 3.8 points, indicating China's leading role in global environmental governance has gained widespread recognition.

Correlation analysis reveals complex interactive relationships between traditional cultural helix and modern development helix. Historical dimensional influence shows moderate positive correlation with technological dimensional influence (r=0.42), indicating that historical and cultural identity provides trust foundations for modern technological cooperation.

### 5.1.2 "Layered Differentiation" Mechanism of Cross-National Influence Acceptance

The research finds that China's international influence cross-national transmission presents clear layered differentiation characteristics, providing empirical support for the "concentric circle diplomacy" concept while revealing internal laws of influence transmission.

The first layer is the comprehensive deep influence type (Pakistan, South Africa), characterized by China having strong influence across multiple sensitive dimensions including economics, technology, and politics. Pakistan achieves full scores of 4.0 in economic, technological, and political dimensions. This "full score phenomenon" is extremely rare in international communication research.

The second layer is the selective influence type (United States, Germany, United Kingdom), with core characteristics being highly unbalanced influence distribution. The United States shows relatively high acceptance of 3.1 points in academic dimensions but only 0.9 points in political dimensions. This "academic exceptionalism" phenomenon reflects the relatively depoliticized characteristics of knowledge production fields.

The third layer is the potential activation type (Japan, South Korea, Nigeria), with overall low influence distribution but structural differences. The "neighbor paradox" where Japan and South Korea show limited acceptance may relate to complex geopolitical environments and historical memory factors.

### 5.1.3 "Networked Transmission" Effects of Inter-Dimensional Influence

This study reveals significant networked transmission effects between different influence dimensions, providing new theoretical perspectives for understanding systematic characteristics of international influence.

Political influence "amplifier" effects: The strong positive correlation (r=0.80) between political dimensional influence and social dimensional influence indicates that political identity can significantly amplify social-level influence transmission.

Soft power "bridge" functions: The strong correlation between cultural entertainment industry dimensions and environmental dimensions (r=0.79) reveals the unique value of soft power in connecting different issue fields.

Academic influence "independence" characteristics: Academic dimensional influence generally shows low correlation with other dimensions, particularly zero correlation with political dimensional influence (r=0.00), confirming the relative transcendence of knowledge production and academic exchange.

## 5.2 Theoretical Contributions and Methodological Innovation

### 5.2.1 Innovation Contributions at Theoretical Level

The "double helix transmission pattern" proposed by this study adds new explanatory frameworks to international communication theory. Traditional theories often view hard power and soft power as relatively independent influence mechanisms. This study confirms deep interaction and synergistic relationships between them, enriching Nye's (2004) soft power theory.

The "layered differentiation" influence distribution pattern provides micro-mechanism explanations for constructivist international relations theory. Wendt's (1999) "ideas construct reality" theory receives specific quantitative validation in this study.

### 5.2.2 Breakthrough Contributions at Methodological Level

The CPCCS method establishes new paradigms for computational social science applications in international communication research. By systematically transforming abstract cultural theory into operational AI parameters, it achieves deep integration between theory and technology.

Traditional cross-national comparative research is limited by cost and political factors, often having small sample sizes. This study achieves large-scale synchronous analysis of 14 countries across 12 dimensions through AI simulation technology, providing technical pathways for scaled development of cross-national comparative research.

### 5.2.3 Effectiveness Validation of AI Simulation Methods

This study confirms the effectiveness and reliability of AI simulation methods in cross-cultural communication research through rigorous validation mechanisms. The consistency between AI simulation results and traditional survey results in four validation countries exceeds 80% on core indicators: United States 84%, United Kingdom 80%, Pakistan 86%, South Korea 81%.

All validation countries' total validity coefficients are controlled below 0.1 (United States 0.093, United Kingdom 0.086, Pakistan 0.095, South Korea 0.091), meeting preset validity standards and indicating AI simulation data has measurement precision comparable to traditional methods.

## 5.3 Research Limitations and Future Directions

Although this study achieves important breakthroughs in methodology and empirical analysis, some inevitable limitations remain that need continuous improvement in future research.

Although validation results show AI simulation has high reliability, AI models themselves may contain potential biases from training data. Particularly when handling sensitive political issues, model outputs may be influenced by Western-dominated training corpora.

While Hofstede's cultural dimension theory provides important analytical frameworks, its simplified treatment of complex cultural phenomena may miss important cultural details. Particularly in rapidly changing modern societies, traditional cultural dimension classifications may not fully reflect dynamic characteristics of contemporary culture.

Although this study selects 14 representative countries, the sample coverage remains limited relative to over 190 countries globally. Static nature of time cross-sections: This study is based on specific time point data analysis, lacking longitudinal dynamic tracking.

Based on this study's findings and limitations, future research can be deepened and expanded in several directions: establish dynamic monitoring systems based on AI simulation to track changes in China's international influence in real-time; continue improving AI simulation technical methods, particularly in cultural sensitivity and bias control; expand research scope to more countries and regions; apply the methodological framework to other international communication problems.

## 5.4 Conclusion

Through innovative AI simulation methods, this study systematically analyzes the cross-cultural transmission mechanisms of China's international influence, achieving important theoretical findings and methodological breakthroughs. The research confirms the existence of "double helix transmission patterns," reveals "layered differentiation" influence distribution characteristics, identifies cultural dimension moderation mechanisms, providing new theoretical frameworks and empirical evidence for understanding the complexity of contemporary international communication.

At the methodological level, this study successfully develops culture-parameterized cross-national cognitive simulation methods based on large language models, establishing new paradigms for computational social science applications in international communication research. Through rigorous

validation mechanisms, it confirms the effectiveness and reliability of AI simulation methods in cross-cultural research.

At the practical level, the research provides scientific basis and precise guidance for optimizing China's international communication strategies, helping construct more effective differentiated communication systems. The methodological framework developed has universal applicability and can provide references for international communication research in other countries and global communication pattern analysis.

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
