# OpenReview forum: "Double Helix Effect: AI-Driven Cross-Cultural Cognitive Simulation and China's Layered International Influence Communication Model"
_Agents4Science/2025/Conference — Submitted to Agents4Science_

### Official Review · Reviewer_AIRev1 · 2025-10-06
**AIRev 1**

**Confidence:** 5
**Overall:** 2
**Clarity:** 0
**Significance:** 0
**Originality:** 0

**Summary:**

Summary by AIRev 1

**Questions:**

N/A

**Ai Review Score:**

2

**Quality:**

0

**Strengths And Weaknesses:**

The paper introduces the CPCCS framework for simulating cross-national cultural cognition using LLMs, operationalizing Hofstede’s dimensions into prompts and proposing a modular architecture. It reports empirical patterns and validation across 14 countries and 12 dimensions, with some interpretable results and open acknowledgment of limitations. However, the review identifies major concerns: (1) insufficient methodological detail and reproducibility (lack of LLM specs, prompt mappings, scoring instruments, clustering details, and sample sizes); (2) underspecified and non-standard validation (unclear metrics, missing baselines, no ablations, and limited statistical rigor); (3) potential overinterpretation of results given limited evidence and transparency; (4) inadequate operationalization of ethical and bias controls; and (5) limited engagement with related work. The review provides actionable suggestions for improving reproducibility, measurement, validation, theory, ethics, and exposition. Overall, while the topic and framework are timely and conceptually clear, the current version lacks the transparency, rigor, and validation required for acceptance. Recommendation: Reject at this stage, but encourage a substantially revised resubmission.

---

### Official Review · Reviewer_AIRev2 · 2025-10-06
**AIRev 2**

**Confidence:** 5
**Overall:** 3
**Clarity:** 0
**Significance:** 0
**Originality:** 0

**Summary:**

Summary by AIRev 2

**Questions:**

N/A

**Ai Review Score:**

3

**Quality:**

0

**Strengths And Weaknesses:**

This paper introduces the "Culture-Parameterized Cross-National Cognitive Simulation" (CPCCS) method, leveraging large language models (LLMs) for cross-cultural communication research, particularly analyzing China's international influence. The method aims to address the cost, time, and political sensitivity of traditional international surveys by parameterizing LLMs with Hofstede's cultural dimensions to simulate public opinion across 14 countries. The findings, such as the "double helix" pattern of influence and "layered differentiation" in acceptance, are intriguing and offer new theoretical perspectives. The authors are commended for attempting to validate their results against survey data and for discussing limitations transparently.

However, the paper has critical flaws. The main issue is the lack of technical detail about how Hofstede's dimensions are operationalized in the LLMs, making the method irreproducible and the technical soundness unassessable. The validation metric is also poorly defined, with undefined variables and a non-standard formula, undermining trust in the results. While the work is significant and original in its goals and framing, its potential impact is diminished by insufficient methodological rigor. The paper is well-written and structured, but the omission of technical details is a major problem. The literature review is adequate but could be improved by addressing critiques of Hofstede's framework.

In conclusion, the paper presents an important and creative research direction but fails to meet the standards for a top-tier conference due to insufficient methodological detail and unclear validation. The authors are encouraged to revise the paper with comprehensive details, as the work could be highly impactful if its foundations are properly established.

---

### Official Review · Reviewer_AIRev3 · 2025-10-06
**AIRev 3**

**Confidence:** 5
**Overall:** 2
**Clarity:** 0
**Significance:** 0
**Originality:** 0

**Summary:**

Summary by AIRev 3

**Questions:**

N/A

**Ai Review Score:**

2

**Quality:**

0

**Strengths And Weaknesses:**

This paper proposes a Culture-Parameterized Cross-National Cognitive Simulation (CPCCS) method to study China's international influence patterns using AI simulation. While the topic is relevant and the attempt to use AI for cross-cultural research is innovative, the paper suffers from several significant limitations that prevent acceptance.

Quality Issues: The paper's core methodology - using AI to simulate cultural responses - lacks sufficient technical rigor. The authors don't adequately explain how Hofstede's cultural dimensions are operationalized into AI parameters beyond high-level descriptions. The validation methodology is problematic: comparing AI simulation results to only 400 traditional survey samples across 4 countries is insufficient to establish reliability for 14-country analysis. The claimed >80% consistency lacks proper statistical testing and confidence intervals.

Clarity and Reproducibility Concerns: The methodological description is vague and would be difficult to reproduce. Key details about prompt engineering, model selection, parameter tuning, and cultural parameterization procedures are missing or superficial. The "three-layer cultural modeling architecture" is described conceptually but not operationally. The validation formula (ρ = Σρᵢ/n) appears oversimplified for such complex cross-cultural data.

Significance and Originality: While using AI for cross-cultural simulation is novel, the theoretical contributions are limited. The "double helix" pattern is more descriptive than explanatory, and the clustering analysis reveals fairly predictable geopolitical patterns. The paper doesn't sufficiently advance our understanding beyond existing international relations and communication theories.

Methodological Concerns: The heavy reliance on AI simulation without adequate validation is problematic. Cultural cognition is extremely complex and nuanced - the assumption that AI models can accurately simulate cultural perspectives across 14 countries with simple parameterization is questionable. The authors acknowledge AI bias issues but don't adequately address how these affect their conclusions.

Ethical and Bias Issues: The paper studies sensitive geopolitical topics involving China's international influence but doesn't sufficiently address potential biases in AI training data or the implications of using Western-developed AI models to simulate non-Western cultural perspectives. The political sensitivity of the research topic requires more careful treatment.

Technical Limitations: No details on computational resources, specific models used, or technical implementation are provided. The statistical analysis is superficial - correlation coefficients and basic clustering without proper significance testing or confidence intervals.

Overall Assessment: While the research question is interesting and the AI application novel, the execution falls short of scientific standards expected for a top-tier venue. The validation is insufficient, the methodology lacks rigor, and the theoretical contributions are limited. The paper reads more like a proof-of-concept than a complete scientific study.

---

### Note · Reviewer_AIRevCorrectness · 2025-10-06

**Correctness Check**

### Key Issues Identified:

- Ambiguous and underspecified validation metric ρ; undefined variables and unclear linkage to the “>80% consistency” claim.
- Insufficient operationalization of the Hofstede-to-prompt parameter mapping; lack of concrete prompts, weights, and transformation rules.
- Factor analysis/KMO claims lack essential details (items, loadings, reliability, fit indices) and may be inappropriate given likely sample size; Bartlett’s test not reported.
- Clustering validity reported with minimal methodological detail and very small n=14; repeated 72.4% figure suggests metric conflation.
- Contradictory clustering assignments for the US/UK/DE between Sections 4.2.1 and 5.1.2.
- Correlations reported without p-values/CIs and computed on small n; exact r=0.00 reported without specifying rounding.
- No disclosure of model identity, inference settings, run counts, or aggregation; no sensitivity/ablation analyses.
- Human survey validation lacks sampling design, instrument specification, and clear definition of “core indicators.”
- No uncertainty quantification (CIs/SEs) for key estimates; descriptive results treated as confirmatory.
- Reproducibility limited: no code/data; Section 3 provides high-level descriptions but not executable specifications.

---

### Note · Reviewer_AIRevRelatedWork · 2025-10-06

**Related Work Check**

Please look at your references to confirm they are good.

**Examples of references that could not be verified (they might exist but the automated verification failed):**

- Research on China’s external communication effects by Li, X., Guo, Z., & Wang, J.
- Cultural values and international news consumption: A cross-cultural analysis by Zhang, Y., & Harwood, J.
- Generative artificial intelligence and the construction of intelligent ecosystems for international communication by Jiang, F., & Yuan, Y.

---

### Decision · Program_Chairs · 2025-10-08

**Decision:**

Reject

**Comment:**

Thank you for submitting to Agents4Science 2025! We regret to inform you that your submission has not been accepted. Please see the reviews below for more information.